# Ice Algae Model Intercomparison Project phase 2 (IAMIP2)

Hakase Hayashida[1,2], Meibing Jin[3,4,5], Nadja S. Steiner[6,7], Neil C. Swart[7], Eiji Watanabe[8], Russell Fiedler[9], Andrew McC. Hogg[2,10], Andrew E. Kiss[2,10], Richard J. Matear[2,9], Peter G. Strutton[1,2]

[1]Institute for Marine and Antarctic Studies, University of Tasmania, Hobart, TAS, Australia
[2]Australian Research Council Centre of Excellence for Climate Extremes, Australia
[3]School of Marine Sciences, Nanjing University of Information Science and Technology, Nanjing, China
[4]Southern Laboratory of Ocean Science and Engineering, Zhuhai, China
[5]International Arctic Research Center, University of Alaska Fairbanks, Fairbanks, AK, USA
[6]Fisheries and Oceans Canada, Institute of Ocean Sciences, Sidney, BC, Canada
[7]Canadian Centre for Climate Modelling and Analysis, Environment and Climate Change Canada, Victoria, BC, Canada
[8]Japan Agency for Marine-Earth Science and Technology, Yokosuka, Kanagawa, Japan
[9]CSIRO Oceans and Atmosphere, Hobart, TAS, Australia
[10]Research School of Earth Sciences, Australian National University, Canberra, ACT, Australia

*Correspondence to*: Hakase Hayashida (hakase.hayashida@utas.edu.au)

**Abstract.** Ice algae play a fundamental role in shaping sea-ice-associated ecosystems and biogeochemistry. This role can be investigated by field observations, however the influence of ice algae at the regional and global scales remains unclear due to limited spatial and temporal coverage of observations, and because ice algae are typically not included in current Earth System Models. To address this knowledge gap, we introduce a new model intercomparison project (MIP), referred to here 20 as the Ice Algae Model Intercomparison Project phase 2 (IAMIP2). IAMIP2 is built upon the experience from its previous phase, and expands its scope to global coverage (both Arctic and Antarctic) and centennial timescales (spanning the mid-twentieth century to the end of the twenty-first century). Participating models are three-dimensional regional and global coupled sea ice–ocean models that incorporate sea-ice ecosystem components. These models are driven by the same initial conditions and atmospheric forcing datasets by incorporating and expanding the protocols of the Ocean Model 25 Intercomparison Project, an endorsed MIP of the Coupled Model Intercomparison Project phase 6 (CMIP6). Doing so provides more robust estimates of model bias and uncertainty, and consequently advances the science of polar marine ecosystems and biogeochemistry. A diagnostic protocol is designed to enhance the reusability of the model data products of IAMIP2. Lastly, the limitations and strengths of IAMIP2 are discussed in the context of prospective research outcomes.

## 1 Introduction

Together with pelagic phytoplankton, microalgae that colonize sea ice are the foundation of polar marine food webs. Understanding the susceptibility of ice algae to climate change is therefore essential to comprehending the climatic impacts on higher trophic levels, such as fish, seals, whales, penguins, polar bears, and humans (Cavan et al., 2019; Darnis et al.,

2012). Vernal blooms of ice algae also directly influence lower trophic levels, such as phytoplankton, zooplankton, and krill, by drawing down near-surface nutrients, reducing light transmission through sea ice, seeding subsequent pelagic blooms, and providing a food source for pelagic and benthic grazers (Leu et al., 2015). Ice algae also regulate ocean biogeochemistry through the biological carbon pump (Mortenson et al., 2020; Watanabe et al., 2015) and polar climates via the production of the climate-active trace gas dimethyl sulfide (Levasseur, 2013) and via the modification of sea-ice albedo (Zeebe et al., 1996).

Field observations of ice algae abundance and distribution are scarce, mainly due to logistical and methodological challenges in these remote and cold environments (Miller et al., 2015), even though technological advancements in recent years have enabled field sampling at much larger scales (Castellani et al., 2020; Cimoli et al., 2020; Lange et al., 2017). Satellite observations are incapable of detecting algae in sea ice. Therefore, the role of ice algae in polar marine ecosystems and biogeochemistry at the regional and global scales remains unclear. One approach to address this knowledge gap is numerical modelling, which simulates ice algae abundance and distribution across the sea-ice domain by incorporating a numerical model of the sea-ice ecosystem into a regional or global three-dimensional coupled sea ice–ocean general circulation model (Vancoppenolle and Tedesco, 2016).

An initial model intercomparison effort was made recently to understand the similarities and differences in simulated ice algae abundance and distribution among the existing three-dimensional models (IAMIP1 hereafter; Watanabe et al., 2019). This model intercomparison investigated the seasonal-to-decadal variability in ice-algal primary productivity in four regions across the Arctic during 1980-2009 simulated by five participating models. The conclusions were: (1) the decadal trend is unclear despite the ongoing reduction of Arctic sea ice; (2) the vernal bloom shifts to an earlier onset and briefer duration over the period of the simulations; and (3) the choice of the maximum growth rate is a key source of the inter-model spread in the simulated ice-algal primary productivity.

Polar regions, especially the Arctic, are warming faster than the rest of the globe, which results in the reduction of both Arctic and Antarctic sea ice (Smith et al., 2019). However, as demonstrated by the previous model intercomparison study (Watanabe et al., 2019), the transient response of ice algae to the anticipated loss of sea ice throughout the twenty-first century may not necessarily be linear. Using a one-dimensional sea-ice biogeochemistry model, Tedesco et al. (2019) found such a non-linear projected response of ice-algal primary productivity to global warming, that also has strong latitudinal dependence. Further investigation using three-dimensional models will provide a comprehensive view of the climate change impacts on polar marine ecosystems and biogeochemistry at the regional-to-global scale, and determine whether ice algae exert any influence on global climate. This knowledge will help to clarify whether ice algae should be incorporated into the next generation of Earth System Models (ESMs), such as those participating in the Coupled Model Intercomparison Project phase 6 (CMIP6; Eyring et al., 2016).

This paper introduces a new model intercomparison effort for ice algae, referred to here as the Ice Algae Model Intercomparison Project phase 2 (IAMIP2). IAMIP2 improves on the previous effort (IAMIP1) in its experimental design, which is based on the experimental protocols of CMIP6. A consistent experimental design allows us to provide more robust estimates of model bias and uncertainty, and consequently advance the science of polar marine ecosystems and biogeochemistry. The scope of IAMIP2 is global, covering both the Arctic and Antarctic, and centennial timescales, spanning the mid-twentieth century to the end of the twenty-first century. IAMIP2 has five main objectives: (1) evaluation of systematic bias in the existing sea-ice ecosystem models; (2) mechanistic understanding of the past changes in ice algae abundance and distribution; (3) assessment of projected changes and their uncertainty in ice algae abundance and distribution; (4) regional-to-global impacts on marine ecosystems and biogeochemistry; and (5) open-access distribution of the model output for sea-ice research and education.

## 2 Participating models

To date, six three-dimensional coupled sea ice–ocean models have been committed to participating in IAMIP2 (Table 1). These models represent either a global or regional configuration of the sea-ice and ocean components of the ESMs or global coupled models (GCMs) participating in CMIP6. A key difference from these parent model components is that the IAMIP2 models include sea-ice ecosystem components that simulate the biological sources and sinks of ice algae biomass and nutrient concentrations in the bottom-ice layer (Fig. 1). In each model, the sea-ice ecosystem component is coupled to the sea-ice component to account for physical processes that regulate the budgets of ice algae biomass and nutrient concentrations, such as sea-ice growth and melt. The sea-ice ecosystem component is also coupled to ocean dynamics and ecosystem components to simulate the tracer exchange at the sea ice–ocean interface. Three of the IAMIP2 models are global configurations, while the other three are pan-Arctic regional configurations in which the tracer states in the ocean are prescribed at the lateral boundaries. The IAMIP2 models are driven by applying a common atmospheric forcing dataset at the surface boundary. We welcome participation of additional models.

### 2.1 ACCESS-OM2

ACCESS-OM2 refers to the sea ice–ocean model of the Australian Community Climate Earth System Simulator (ACCESS), and is described in detail in Kiss et al. (2020). In brief, ACCESS-OM2 consists of an ocean dynamics component based on the Modular Ocean Model (MOM) version 5.1 and a sea-ice dynamics component based on the Los Alamos sea ice model (CICE) version 5.1.2. Notably, these sea ice–ocean physical model components are identical to those adopted for ACCESS-CM2, the GCM of ACCESS contributing to CMIP6 (Bi et al., 2020).

For IAMIP2, ocean and sea-ice ecosystem components are added to ACCESS-OM2. The ocean ecosystem component is the Whole Ocean Model of Biogeochemistry and Trophic-dynamics (WOMBAT), which consists of nitrate, iron, phytoplankton, zooplankton, detritus, dissolved oxygen, dissolved inorganic carbon, total alkalinity, and calcium carbonate (Ziehn et al.,

2020). The sea-ice ecosystem component is the Biogeochemistry of CICE (Jeffery et al., 2016), which is based on Jin et al. (2006). For IAMIP2, ACCESS-OM2 simulates ice algae biomass and nitrate concentration in the bottom-ice layer, which are coupled to phytoplankton biomass and nitrate concentration in the ocean surface layer, respectively. The horizontal resolution of ACCESS-OM2 is nominally 1° ($360 \times 300$) and the vertical resolution in the ocean surface layer is 2.3 m (Kiss et al., 2020).

## 2.2 CESM-IARC

CESM-IARC refers to the 1° global sea ice–ocean configuration of the Community Earth System Model (CESM) version 1 with a sub-grid-scale brine rejection parameterization that improves ocean mixing under sea ice (Jin et al., 2018). The ocean dynamic component of CESM-IARC is the Parallel Ocean Program (POP) version 2 and the sea-ice dynamic component is CICE version 5.1.2. The vertical resolution of the ocean surface layer is 10 m. The ocean ecosystem component consists of

multiple nutrients (nitrate, ammonium, iron, silicate, and phosphate), phytoplankton functional groups (diatoms, flagellates, and diazotrophs), a single zooplankton group, dissolved organic matter (nitrogen, carbon, iron, and phosphorus), dissolved oxygen, dissolved inorganic carbon, and total alkalinity (Moore et al., 2013). The sea-ice ecosystem component consists of nitrate, ammonium, silicate, and ice algae (Jin et al., 2006).

## 2.3 RASM

RASM refers to the 1/12° sea ice–ocean configuration of the Regional Arctic System Model, and consists of the same physical and ecosystem model components as CESM-IARC. The spatial coverage of RASM is pan-Arctic with lateral boundaries located approximately along the 40 °N Atlantic sector and along the 30 °N Pacific sector (Jin et al., 2018). The vertical resolution of the ocean surface layer is 5 m.

## 2.4 CanNEMO

CanNEMO is the ocean component of CanESM5 (Swart et al., 2019), which is modified from version 3.4.1 of the Nucleus for European Modelling of the Ocean (NEMO; Madec and the NEMO team, 2012), and includes the Louvain-la-Neuve sea Ice Model version 2 (LIM2; Bouillon et al., 2009; Fichefet and Maqueda, 1997). The modifications include the addition of a lee wave mixing scheme based on Saenko et al. (2012) and an update to the mesoscale eddy mixing length-scale (Saenko et al., 2018), as well as various parameter settings, as described in Swart et al. (2019). CanNEMO is configured on the ORCA1

tripolar grid, with a nominal grid spacing of 1°, refining to 1/3° within 20° of the equator. There are 45 vertical levels, ranging from 6 m near the surface to 250 m in the abyss. In the version used here, ocean biogeochemistry is represented by the Canadian Ocean Ecosystem (CanOE) model. CanOE contains two classes of phytoplankton, zooplankton and detritus,

with variable elemental (C/N/Fe) ratios in phytoplankton and fixed ratios for zooplankton and detritus; as well as prognostic carbonate chemistry, and cycles of iron, calcium carbonate and parameterized nitrogen fixation and denitrification (Swart et al., 2019).

For IAMIP2, CanNEMO–CanOE is coupled to the Canadian Sea-Ice Biogeochemistry (CSIB) model, which consists of nitrate, ammonium, and ice algae (Hayashida et al., 2019). Modelled ice algae have a higher sensitivity to low-light conditions relative to modelled phytoplankton (Mortenson et al., 2017). When modelled ice algae are released from sea ice into seawater, they contribute partly to the large detritus pool and the seeding of large phytoplankton.

## 2.5 NEMO-NAA

NEMO-NAA is a pan-Arctic regional sea ice–ocean model based on NEMO version 3.4 coupled to LIM2 (Hu and Myers, 2013). The version used here incorporates several modifications that improve the simulation of physical and biogeochemical processes in the Arctic (Hayashida et al., 2019; in revision). Specifically, these modifications include light penetration through snow and sea ice, the vertical resolution of the ocean model, river runoff of biogeochemical tracers, removal of iron dependency of phytoplankton and zooplankton, and changes to sea-ice model parameters. The horizontal grid resolution is 1/4°, which ranges from 10 km near the North American coastline to 14.5 km along the northern Eurasian coastline. The vertical resolution ranges from 1 m for the upper surface layer to 255 m in the deep ocean. Similar to CanNEMO, the ecosystem component of NEMO-NAA is CSIB–CanOE.

## 2.6 COCO–Arctic NEMURO

COCO–Arctic NEMURO refers to a pan-Arctic regional sea ice–ocean model developed at the Japan Agency for Marine-Earth Science and Technology (JAMSTEC; Watanabe et al., 2019). The physical model component is the Center for Climate System Research Ocean Component Model (COCO) version 4.9 (Hasumi, 2006). The sea-ice component of COCO accounts for seven-category distributions of sub-grid snow depth and ice thickness with a one-layer thermodynamic formulation (Bitz and Lipscomb, 1999; Bitz et al., 2001; Lipscomb, 2001) and the elastic–viscous–plastic rheology (Hunke and Dukowicz, 1997). The model domain covers the entire Arctic Ocean with boundaries at the North Atlantic (45 °N) and at the Bering Strait on the Pacific side. The Bering Strait throughflow across the Pacific model boundary is prescribed by idealized seasonal cycles of temperature, salinity, and current velocity based on Woodgate et al. (2005). The model adopts the spherical coordinate system rotated by 90°, which sets the singular points (the North and South Poles of the model grid) at the equator. The horizontal resolution of COCO–Arctic NEMURO is 1/4°, and there are 28 vertical levels in the ocean with variable resolutions from 2 m in the uppermost layer to 500 m below 1,000 m depth. The ecosystem model component is the Arctic and North Pacific Ecosystem Model for Understanding Regional Oceanography (Arctic NEMURO; Watanabe et al., 2015). The ocean ecosystem model component consists of 3 nutrients (nitrate, ammonium, and silicate), 5 plankton types (diatoms, flagellates, microzooplankton, copepod, and predator zooplankton), dissolved organic nitrogen, particulate organic

nitrogen, and opal (Kishi et al., 2007). The sea-ice ecosystem model component consists of ice algae, ice-related fauna, and particulate organic matter (Watanabe et al., 2015).

## 3 Experiments

The experimental design of IAMIP2 is developed based on the experience from IAMIP1 and the Ocean Model Intercomparison Project (OMIP; Griffies et al., 2016; Orr et al., 2017), which is an endorsed Model Intercomparison Project (MIP) of CMIP6. The OMIP protocol is useful because OMIP is based on sea ice–ocean models driven by common atmospheric forcing fields at their surface boundaries. Applying common forcing eliminates uncertainties due to atmospheric processes and feedbacks, and allows us to focus on differences in the sea-ice, ocean, and biogeochemistry components.

Four numerical experiments are planned for IAMIP2: historical, projection, exclusion, and control as described in detail below and shown in Fig. 2. The IAMIP2 models are expected to conduct all these experiments.

### 3.1 Historical

The historical experiment is designed to simulate changes in ice algae abundance and distribution since the mid-twentieth century, the period for which we have realistic surface-atmospheric conditions based on observations. This experiment spans the 61 years from January 1, 1958, to December 31, 2018, and uses the Japanese 55-year atmospheric reanalysis for driving sea ice–ocean models version 1.4.0, referred to here as JRA55-do (Tsujino et al., 2018). For reference, version 1.4.0 is also used for the second phase of OMIP (Tsujino et al., 2020). JRA55-do is regarded as a successor of the Coordinated Ocean-ice Reference Experiment version 2 forcing dataset (CORE2; Large & Yeager, 2009), and has finer spatial and temporal resolutions than CORE2. The surface atmospheric variables of JRA55-do are air temperature at 10 m, specific humidity at 10 m, eastward wind at 10 m, northward wind at 10 m, sea level pressure, downward shortwave radiation, downward longwave radiation, rainfall flux, snowfall flux, river runoff, and calving flux (Fig. 1). The original spatial and temporal resolutions of these variables are respectively ~0.5° and 3 hourly except for river runoff and calving flux, which are provided at 0.25° and daily (Tsujino et al., 2018). However, in reality, the daily calving flux is provided by linear interpolation of monthly data for Greenland and it is temporally constant for Antarctica (Tsujino et al., 2018). Although JRA55-do provides calving flux, none of the IAMIP2 models have iceberg components, and therefore it is released into the ocean as meltwater, adopting the approach before version 1.4.0 (Tsujino et al., 2018). JRA55-do is interpolated to the model grid either prior to or during the experiment.

All IAMIP2 models are initialized from rest (three-dimensional oceanic velocity fields and two-dimensional sea level fields are all set to zero in the first time step) and with ocean temperature, salinity, dissolved oxygen, and nutrients (nitrate, phosphate, and silicate) from the World Ocean Atlas version 2 (WOA13v2; Garcia et al., 2013; Locarnini et al., 2013;

Zweng et al., 2013). More specifically, these fields are the version of WOA13v2 provided for OMIP (Griffies et al., 2016; Orr et al., 2017). There is no recommended protocol for the initialization of sea-ice and other biogeochemical fields. As discussed in Section 2.3 of Griffies et al. (2016), the restoring of sea-surface salinity is necessary to reduce drift in sea ice–ocean models over decadal timescales. However, there is no best practice for salinity restoring because it depends on model details. Therefore, the restoring procedure is left to the discretion of the participating groups, although it is recommended to choose a weak restoring as much as possible to minimize the impact on variability.

Three of the IAMIP2 models are based on pan-Arctic regional configurations that require lateral boundary conditions. How to prescribe these conditions is left to the discretion of the participating groups.

Although carbonate chemistry is not the primary focus of IAMIP2, we recommend that modelled total alkalinity and dissolved inorganic carbon should be initialized with the Global Ocean Data Analysis Project version 2 (GLODAPv2; Lauvset et al., 2016) and prescribe the monthly global-mean atmospheric carbon dioxide concentration for the air-sea carbon flux (Meinshausen et al., 2017). These procedures are consistent with OMIP (Orr et al., 2017), and allow us to expand the use of the IAMIP2 product for future research.

### 3.2 Projection

The projection experiments are designed to simulate the projected changes in ice algae abundance and distribution throughout the twenty-first century under two of the greenhouse gas emission scenarios for CMIP6, known as the Shared Socioeconomic Pathways 1-2.6 and 5-8.5 (SSP1-2.6 and SSP5-8.5; O'Neill et al., 2016). SSP1-2.6 is a low emission scenario that informs the Paris Agreement goal of limiting the global warming to below 2 °C of the pre-industrial level. SSP5-8.5 is the highest emission scenario of CMIP6. Therefore, conducting these projections allows us to assess and compare between the impacts of strong mitigation and fossil-fueled development (O'Neill et al., 2016). Each of these experiments spans 86 years from 2015 to 2100. The IAMIP2 models are initialized from states at the end of 2014 in the historical experiment. The atmospheric carbon dioxide concentrations are set to their monthly global-mean values prescribed for its respective SSP (Meinshausen et al., 2020).

### 3.2.1 Selection of the projected atmospheric forcing dataset

The IAMIP2 models are driven by the atmospheric output of selected CMIP6 models that provide the atmospheric forcing fields at the nominal spatial resolution of 100 km and at the temporal resolutions needed for simulating high-frequency (e.g., daily) variability (Holdsworth and Myers, 2015; Lebeaupin Brossier et al., 2012). To date, there are four CMIP6 models that satisfy these criteria: CMCC-CM2-SR5, CMCC-ESM2, EC-Earth3, and MRI-ESM2-0. Specifically, these models provide the simulated atmospheric forcing variables over 2015-2100 under both SSP1-2.6 and SSP5-8.5 at the temporal resolutions identical to those of JRA55-do except for sea level pressure, river runoff, and calving flux. Sea level pressure fields are

provided 6 hourly, as opposed to 3 hourly in JRA55-do. River runoff fields are provided at monthly, as opposed to daily in JRA55-do. Calving flux fields are not provided at all under any of the SSP scenarios. To partly overcome this limitation, we prescribe a monthly climatology of the calving flux of JRA55-do over 1958-2018 for the projection experiment. This will provide interannually-invariant calving flux for Greenland, and constant calving flux for Antarctica as noted in Section 3.1.

Among the four CMIP6 models, we choose the output of CMCC-ESM2 and EC-Earth3 for the atmospheric forcing for the projection experiments because CMCC-ESM2 and EC-Earth3 provide overall the most realistic atmospheric conditions (in best agreement with JRA55-do) for the South and North polar oceans, respectively. Specifically, we compare the global and polar surface air temperature and major climate modes derived from JRA55-do with those simulated by the following 26 CMIP6 models over 1958-2100 (Figures 3 and 4): ACCESS-CM2, ACCESS-ESM1-5, AWI-CM-1-1-MR, BCC-CSM2-MR, CESM2-WACCM, CIESM, CMCC-CM2-SR5, CMCC-ESM2, CanESM5, EC-Earth3, EC-Earth3-Veg, FGOALS-f3-L, FGOALS-g3, FIO-ESM-2-0, GFDL-ESM4, INM-CM4-8, INM-CM5-0, IPSL-CM6A-LR, KACE-1-0-G, MIROC6, MPI-ESM1-2-HR, MPI-ESM1-2-LR, MRI-ESM2-0, NESM3, NorESM2-LM, and NorESM2-MM. Although the objective here is to compare among the four candidates for the projected atmospheric forcing, we include the remaining 22 CMIP6 models that have provided the output at monthly resolution as well as the multi-model mean product of the 26 models for reference. These model products belong to Variant Label *r1i1p1f1* of Experiment IDs *historical* and *ssp585*, and they were downloaded from https://esgf-node.llnl.gov/ (accessed on July 15, 2021).

### 3.2.2 Comparison of surface air temperature over the ocean

Surface air temperatures are averaged over the ocean grid cells globally, north of 60 °N, and south of 60 °S (Figure 3). In terms of global averages, EC-Earth3 simulates surface air temperature closest to JRA55-do prior to 2000 (Figure 3a). After 2000, EC-Earth3 simulates warmer surface air temperature that agrees well with CMCC-CM2-SR5 but deviates from JRA55-do. At the beginning of the SSP projections (2015), the global average of CMCC-ESM2 matches well with JRA55-do. MRI-ESM2-0 consistently underestimates the global averages. Compared with the multi-model averages, EC-Earth3 projects greater global warming especially in the latter half of the 21$^{st}$ century, whereas MRI-ESM2-0 projecton is close to the multi-model averages (Figure 3b).

In terms of Arctic averages, MRI-ESM2-0 simulates surface air temperature that is in closest agreement with JRA55-do, but EC-Earth3 compares well later in the 2010s (Figure 3c). In 2015, EC-Earth3 is closest to JRA55-do among the four candidates. Polar amplification is evident in all models, but is more so in EC-Earth3 (>15 °C warming by the end of the 21$^{st}$ century relative to the 1958-2014 average; Figure 3d).

In the Antarctic, the average air temperatures of CMCC-CM2-SR5 and CMCC-ESM2 agree by far the best with JRA55-do compared to the other two candidates (Figure 3e). EC-Earth3 simulates roughly 4 °C warmer than JRA55-do, whereas MRI-ESM2-0 simulates approximately 3 °C colder surface air over the South polar ocean. MRI-ESM2-0 exhibits greater interannual variability throughout the 21st century and the greatest Antarctic warming among the four candidates (approximately 5 °C warming by 2100 relative to the 1958-2014 average; Figure 3f).

### 3.2.3 Comparison of major climate modes of variability

Three major modes of climate variability considered here are the El Niño–Southern Oscillation (ENSO), the Northern Annular Mode (NAM), and the Southern Annular Mode (SAM). ENSO is the dominant mode of climate variability for the globe, which can be characterised by an index called the Equatorial Southern Oscillation Index (EQSOI; Bell and Halpert, 1998). EQSOI is the standardised sea level pressure difference between two regions over Indonesia (80 °W-130 °W, 5 °S-5 °N) and the eastern equatorial Pacific (90 °E-140 °E, 5 °S-5 °N). Because the CMIP6 models have their own ENSO variability, EQSOIs of these models are not expected to be in phase with that of JRA55-do (this is applicable also to other modes of climate variability, such as NAM and SAM discussed below). Instead, we compare the magnitude of the sea level pressure difference, which determines the strength of the easterly winds along the equator (Figure 4a). Furthermore, we compare the power spectra of EQSOI over 1958-2014 (Figure 4b).

CMCC-CM2-SR5 and CMCC-ESM2 simulate the sea level pressure difference for EQSOI that is in closer agreement with JRA55-do than the other two candidates whose sea level pressure difference is about 100 Pa lower than JRA55-do (Figure 4a). JRA55-do shows two maxima in the spectrum of EQSOI at periods between 3 and 6 years (Figure 4b). MRI-ESM2-0 captures the one of these maxima at period of about 3-4 years, while CMCC-ESM2 exhibits the other maximum at about 5-6 years. EC-Earth3 shows a maximum at period of 4-5 years. In contrast, the EQSOI signal of CMCC-CM2-SR5 is peaked at a longer period of 7-8 years.

NAM is the dominant mode of climate variability in the Northern Hemisphere, which can be characterised by the sea level pressure difference between the zonal means at 35 °N and 65 °N (Li and Wang, 2003). JRA55-do overall exhibits smaller sea level pressure difference than any of the four candidates or the multi-model averages (Figure 4c). MRI-ESM2-0 simulates somewhat larger sea level pressure difference than the other three candidates that are close to the multi-model averages. The power spectra of NAM are noisier at higher-frequency (<1 year) and the peaks are spread more broadly than those of EQSOI among the candidates and JRA55-do (Figure 4d). JRA55-do shows peaks at about 0.5, 1, and 10 years, indicating the importance of seasonal, interannual, and interdecadal variability of NAM, respectively. The four candidates also exhibit peaks within this seasonal-to-interdecadal time scale, but differ in terms of the exact locations of these peaks. The strongest signal of NAM is present at period of roughly 0.5-1 year (MRI-ESM2-0), 3 years (CMCC-CM2-SR5 and CMCC-ESM2), and 4-5 years (EC-Earth3).

SAM is the dominant mode of climate variability in the Southern Hemisphere, which can be characterised by the sea level
pressure difference between the zonal means at 40 °S and 65 °S (Gong and Wang, 1999). EC-Earth3 and MRI-ESM2-0
compare equally well with JRA55-do in simulating the sea level pressure difference for SAM, which is lower than the multi-
model mean of the 26 CMIP6 models (Figure 4e). In contrast, CMCC-CM2-SR5 and CMCC-ESM2 simulate about 500 Pa
greater sea level pressure difference than JRA55-do. Similar to NAM, the power spectra of SAM are distributed across the
broad time scales, but the peaks occur at periods of about 1 year or greater (Figure 4f). The only exception is EC-Earth3, in
which the strongest signal is present at period of slightly less than 1 year. CMCC-CM2-SR5 agrees well with JRA55-do in
simulating the absolute maximum at period of 1-2 years. Consistent among the four candidates and JRA55-do, SAM exhibits
appreciable variability also at lower frequencies, as indicated by the presence of local maxima at periods of 3-6 years and
beyond 10 years. Notably, this low-frequency variability is the strongest signature of SAM simulated by MRI-ESM2-0.

### 3.2.4 Recommendations for the projected atmospheric forcing dataset

In summary, all four candidates for the projected atmospheric forcing dataset simulate reasonable historical atmospheric
conditions compared to JRA55-do. An exception is the substantially colder surface air temperature over the Antarctic
oceanic region (Figure 3e), for which CMCC-CM2-SR5 and CMCC-ESM2 are superior to the other two models. In contrast,
the latter two models outperform the former in simulating surface air temperature over the Arctic oceanic region (Figure 3c).
Therefore, there is no single candidate that does a better job than the others at both poles. For this reason, we adopt the
output of two models, CMCC-ESM2-0 and EC-Earth3, as the standard atmospheric forcing dataset for the projection
experiments of IAMIP2. With two SSP scenarios, there are four combinations (CMCC-ESM2-0 ssp126, CMCC-ESM2-0
ssp585, EC-Earth3 ssp126, and EC-Earth3 ssp585; Table 2). We encourage the IAMIP2 participants to perform all the four
projections. However, if computational resources are limited to undertake only one projection, we request one of the SSP5-
8.5 projections, either CMCC-ESM2-0 ssp858 (for Antarctic-focused models) or EC-Earth3 ssp585 (for Arctic-focused
models). Depending on computational resources, additional projection experiments using the output of the other two
candidates as well as other CMIP6 models may be considered if they provide the output at high temporal resolution. Such
additional experiments will allow assessment of the sensitivity of the IAMIP2 models to projected atmospheric conditions.

### 3.3 Exclusion

The exclusion experiment is designed to simulate ocean biogeochemistry in the absence of ice algae, which is the case for all
CMIP6 models. This experiment is done in the same set up as the historical experiment except that the sea-ice ecosystem
component is excluded. Comparing the results of the historical and exclusion experiments allows us to quantify the impacts
of ice algae on polar marine lower-trophic-level ecosystems and the biological carbon pump. Doing so assesses the
significance of incorporating sea-ice ecosystem components into the next-generation ESMs.

## 3.4 Control

The control experiment is designed to diagnose artificial model drifts and to distinguish anthropogenic effects from natural variability. This experiment spans 143 years by repeating the annual cycle of JRA55-do from May 1, 1990 to April 30, 1991, during which major climate modes were all in neutral phases (Stewart et al., 2020).

## 4 Diagnostics

The following guidelines are developed for IAMIP2 in order to implement the Findability, Accessibility, Interoperability,
and Reusability (FAIR) data principles (Wilkinson et al., 2016) and ensure that the IAMIP2 product reaches the end users (Objective 5). Specifically, we make the product discoverable from the website http://cosima.org.au/index.php/working-groups/iamip2 which provides a list of hyperlinks to the IAMIP2 product archived by individual modelling groups.

All model diagnostics are saved as daily averages in the NetCDF format, and where applicable, their names must follow the
CMIP6 naming conventions (Table 2). Daily temporal resolution is needed to quantify the bloom phenology (Watanabe et al., 2019) and to provide the ocean and sea-ice climate data for driving sea-ice biogeochemical models (Lavoie et al., 2010; Tedesco et al., 2019). To limit data storage needs, only two-dimensional fields are saved, and to preserve the spatial details in regional models, they are stored on the models' native grids. To perform interpolation for analysis, an additional file containing longitude, latitude, and grid-cell area needs to be provided for each model. The output should be stored using a
common format for directory and file names (Figure 5).

A few examples of potential use of these diagnostics for investigating various roles of ice algae are illustrated here. Their ecological role as the foundation of the polar marine food web and their relative importance can be quantified by comparing the biomass (*phycbi* and *phycos*; Table 2) and primary productivity between ice algae and phytoplankton (*intppbi* and *intpp*).
The latter quantities can also be used to quantify the biogeochemical role of ice algae in carbon fixation and their contribution to the biological carbon pump can be assessed using particulate organic carbon export (*epc100*). A combination of physical (*siconc*, *paros*, *sst*, and *mlots2t*) and biogeochemical diagnostics (*no3os* and *phyos*) can be used to estimate dimethyl sulfide concentration (e.g., Bock et al., 2021; Galí et al., 2019) as well as its emission using the wind speed which is available as the atmospheric forcing fields (*uas* and *vas*; Figure 1).

## 345 5 Discussion

Although the experimental design of IAMIP2 advances from that of IAMIP1 (Watanabe et al., 2019) in many aspects, it has its own limitations which are discussed here to help the interpretation of results and for consideration by prospective end users. First, the vertical extent of the sea-ice ecosystem component of the currently committed selection of IAMIP2 models

is restricted to the bottom-ice layer, which likely underestimates depth-integrated ice algae biomass and primary production because ice algae are present throughout the sea-ice column especially in the Antarctic. Although simulating ice algae throughout the sea-ice column is desirable and has been conducted under one-dimensional settings (Duarte et al., 2015; Pogson et al., 2011) as well as in a recent study using an ESM (Jeffery et al., 2020), its implementation into high-resolution three-dimensional models is computationally expensive. We anticipate that this limitation has a negligible effect on the estimation of depth-integrated biomass and primary production in landfast sea ice (Meiners et al., 2018), but it could underestimate these quantities substantially for pack ice as demonstrated by field observations (Meiners et al., 2012). We consider this a necessary compromise in order to progress IAMIP2 with currently-available computational resources.

For projections of sea ice-ocean models, a few previous studies have prescribed a synthetic atmospheric forcing dataset instead of applying the raw output of climate models in order to ensure that there is no undesirable step change in the atmospheric conditions at the beginning of projections (Naughten et al., 2018) and that the high-frequency climate variability is simulated realistically (Zhang et al., 2017). In contrast to the approach taken by these previous studies, the IAMIP2 models are driven by the raw output of CMIP6 model projections, which is made possible thanks to four CMIP6 models so far that have provided their atmospheric output available at the temporal resolutions needed to simulate the high-frequency climate variability (Holdsworth and Myers, 2015; Lebeaupin Brossier et al., 2012). Our approach is advantageous over that of the previous studies in that the IAMIP2 model projections account for projected changes in both the high- and low-frequency climate variability. As demonstrated in Section 3.2, there appears to be no single CMIP6 model that outperforms the other models considered here in simulating surface air temperature in both marine polar regions. Some models are better at simulating Arctic surface air temperature but not the Antarctic, and vice versa. Considering this finding, we aim to perform multiple projections by prescribing the atmospheric output of two CMIP6 models (EC-Earth3 and CMCC-ESM2) under two emission scenarios (SSP1-2.6 and SSP5-8.5). To a certain extent, these projections will allow us to investigate the uncertainty in the projected changes in ice algae abundance and distribution due to the uncertainty in the projected atmospheric conditions under the same climate change scenarios. Furthermore, the implications of climate change scenarios for polar marine ecosystems and biogeochemistry can be addressed by comparison of the SSP1-2.6 and SSP5-8.5 projections.

One important source of biases in IAMIP2 projections by regional models is the lateral boundary conditions for nutrients. As a limiting factor for primary production in sea ice, prescribing the temporally-varying nutrient boundary conditions are desirable. However, this is not an easy task given the large uncertainty in projected changes in nutrients especially in polar regions (e.g., Lannuzel et al., 2020).

IAMIP2 is an ongoing international effort aiming primarily to understand the role of ice algae in polar marine ecosystems and biogeochemistry at the regional and hemispheric scales. This paper describes the design of IAMIP2 which is built upon

the experience from IAMIP1 (Watanabe et al., 2019) and by keeping up to date with the CMIP6 protocols (Griffies et al., 2016; O'Neill et al., 2016; Orr et al., 2017). Six three-dimensional regional and global coupled sea ice–ocean models are currently committed to participating in IAMIP2. These models are driven by the same initial conditions and atmospheric forcing dataset to assess systematic biases in these models in terms of simulating ice algae abundance and distribution. Five numerical experiments are designed to understand the past changes since the mid-twentieth century by applying realistic atmospheric forcing (JRA55-do) as well as the projected changes throughout the twenty-first century by applying the high-temporal-resolution output of selected CMIP6 model projections under the SSP1-2.6 and SSP5-8.5 scenarios. Two other experiments are used to separate the anthropogenic effect from natural variability and quantify the large-scale impacts of incorporating ice algae into regional and global models on polar marine ecosystems and biogeochemistry. The model data products of IAMIP2 are expected to meet the FAIR data principles and intended to be used for future research and as educational tools. In conclusion, IAMIP2 is expected to advance the science of polar marine ecosystems and biogeochemistry.

**Data and code availability**

The model output of IAMIP2 will be made available by individual modelling groups, which will be discoverable from the website http://cosima.org.au/index.php/working-groups/iamip2. The atmospheric forcing datasets for the four projection experiments are available on the National Computational Infrastructure (NCI) National Research Data Collection (Hayashida, 2021).

**Author contributions**

HH conceived IAMIP2 and developed the details with MJ, NSS, NCS, and EW. AH, AK, RM, and PS supervised the ACCESS-OM2 contribution to IAMIP2. RF and HH developed the model code for enabling sea ice-ocean coupling of biogeochemistry in ACCESS-OM2. HH prepared the projected atmospheric forcing datasets, performed the analysis, and wrote the paper with input from all co-authors.

**Competing interests**

All authors declare that they have no conflict of interest.

**Acknowledgements**

HH thanks the members of the Consortium for Ocean-Sea Ice Modelling in Australia (COSIMA) for discussion on atmospheric forcing dataset for future projections. This dataset is made available through technical support from Paola

Petrelli from the CMS team of the Australian Research Council Centre of Excellence for Climate Extremes (CLEX). HH, AH, AK, RM, and PS acknowledge support from CLEX (CE170100023). This study was undertaken with the assistance of resources and services from the NCI, which is supported by the Australian Government. Analyses are performed using the Climate Data Operator (CDO) software (Schulzweida, 2019) and Python. Normalised power spectra are computed using SciPy (Virtanen et al., 2020). EW is supported by the Grant-in-Aid for Scientific Research of Japan Society for the Promotion of Science (JSPS) (KAKENHI 18H03368) and the Arctic Challenge for Sustainability II (ArCS II) project (JPMXD1420318865) in Japan. NSS and NCS acknowledge funding through the Departments of Environment and Climate Change Canada and Fisheries and Oceans Canada. This work is a contribution to the Biogeochemical Exchange processes at Sea Ice Interfaces (BEPSII) network.

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

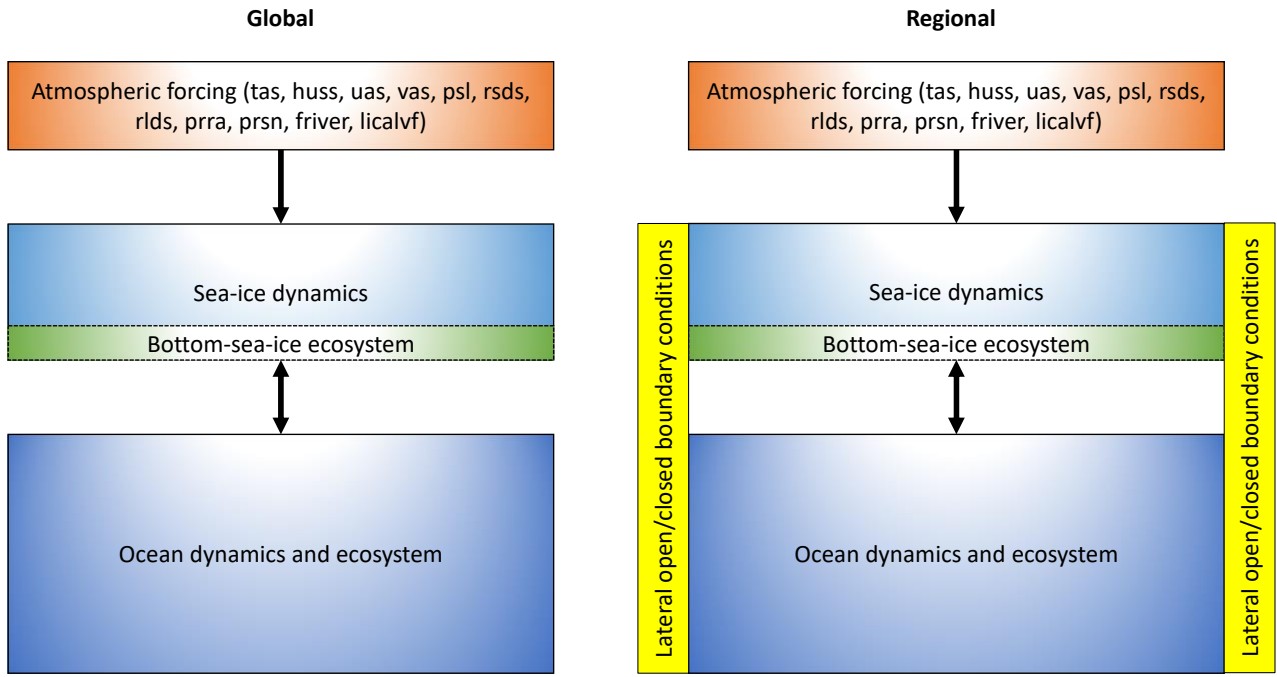

**Figure 1: Global (left) and regional (right) configurations of the three-dimensional coupled sea ice–ocean physical–biogeochemical models participating in IAMIP2.** The surface-atmospheric forcing dataset consists of air temperature at 10 m (tas), specific humidity at

10 m (huss), eastward wind at 10 m (uas), northward wind at 10 m (vas), sea level pressure (psl), downward shortwave radiation (rsds), downward longwave radiation (rlds), rainfall flux (prra), snowfall flux (prsn), river runoff (friver), and calving flux (licalvf).

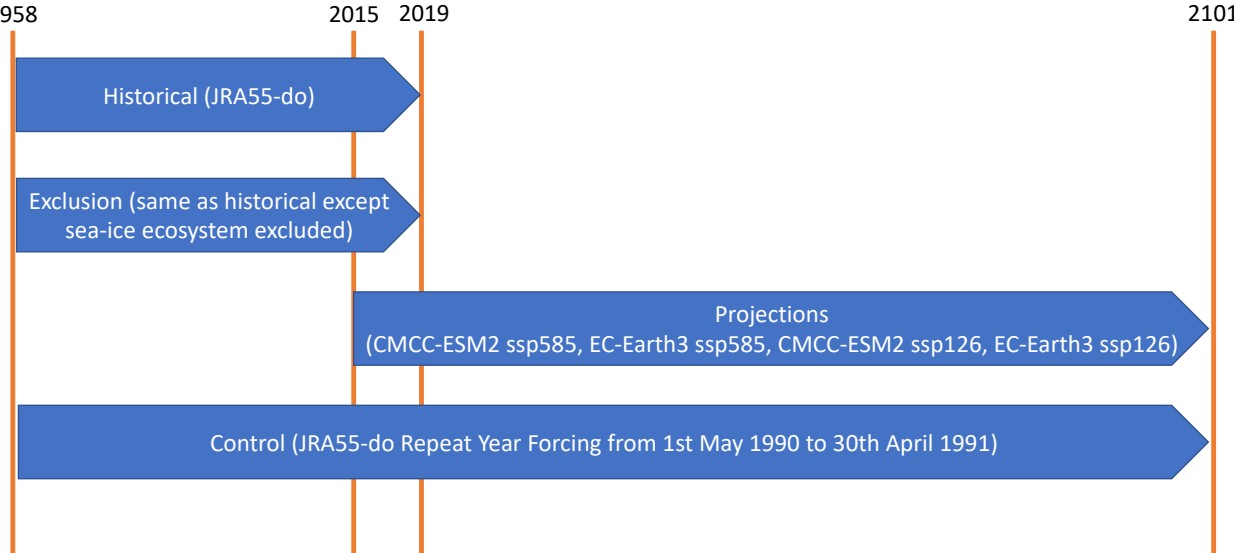

600     **Figure 2: Timeline of the numerical experiments of IAMIP2.** Years begin on January 1. Each experiment starts on January 1 and ends on December 31. For example, the historical experiment starts from January 1, 1958, and ends on December 31, 2018.

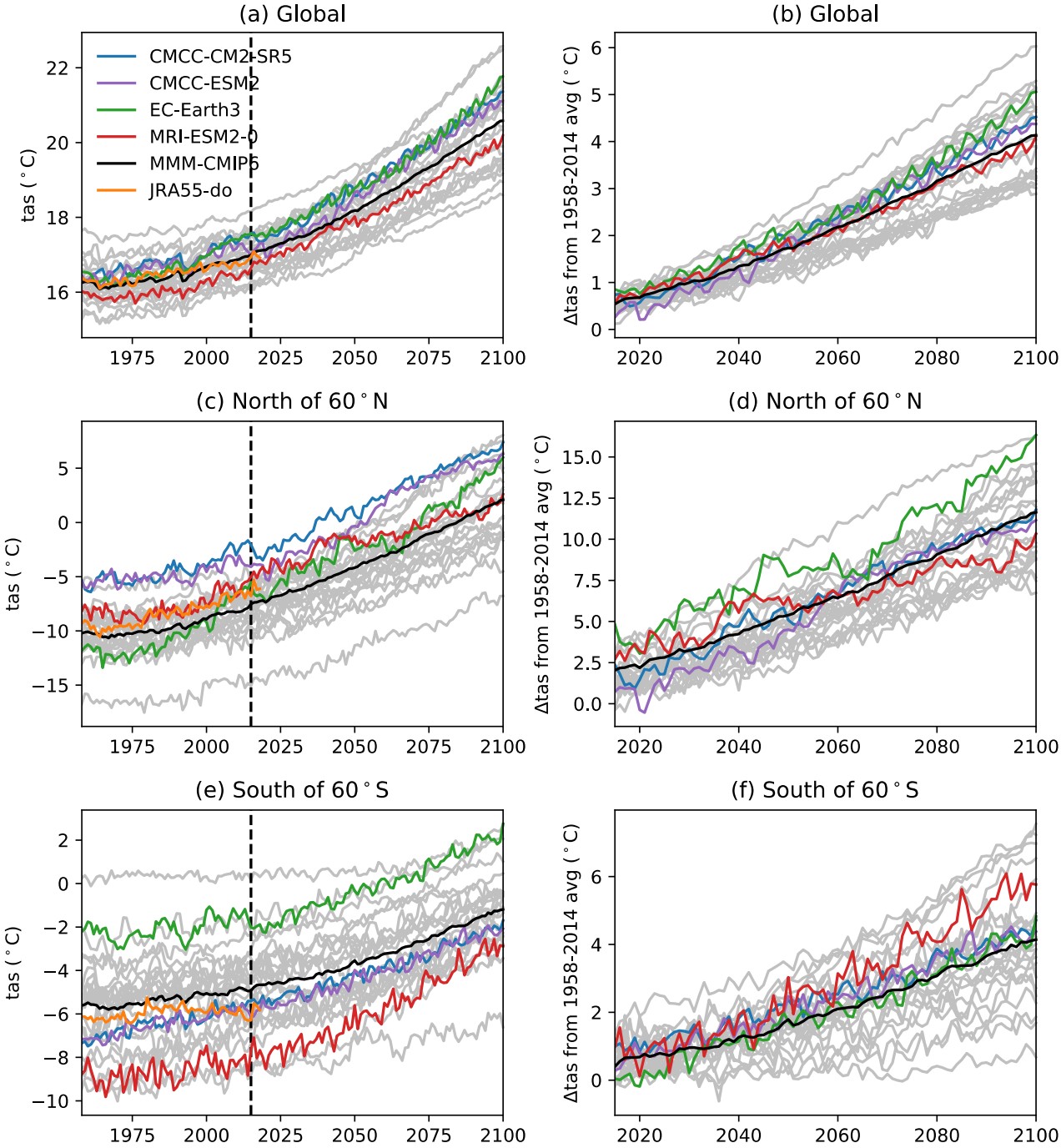

**Figure 3: Comparisons of historical and projected global and polar warming simulated by 26 CMIP6 models over the historical period (1958-2014) and the projection period under the SSP5-8.5 scenario (2015-2100).** Time series of annual mean surface air temperature averaged over the ocean grid cells of (a) the globe, (c) the Northern Hemisphere north of 60 °N, and (e) the Southern Hemisphere south of 60 °S. Projected changes in annual mean surface air temperature averaged over the ocean grid cells of (b) the globe,

(d) the Northern Hemisphere north of 60 °N, and (f) the Southern Hemisphere south of 60 °S, relative to their averages during 1958-2014. Blue, purple, green, and red denote 4 of the 26 models that provide the atmospheric output at the temporal resolutions needed for the projection experiments of IAMIP2. Gray denotes the remaining 22 models. Black denotes the multi-model mean of the 26 models. Orange denotes the JRA55-do dataset used for the atmospheric forcing for the historical experiment of IAMIP2.

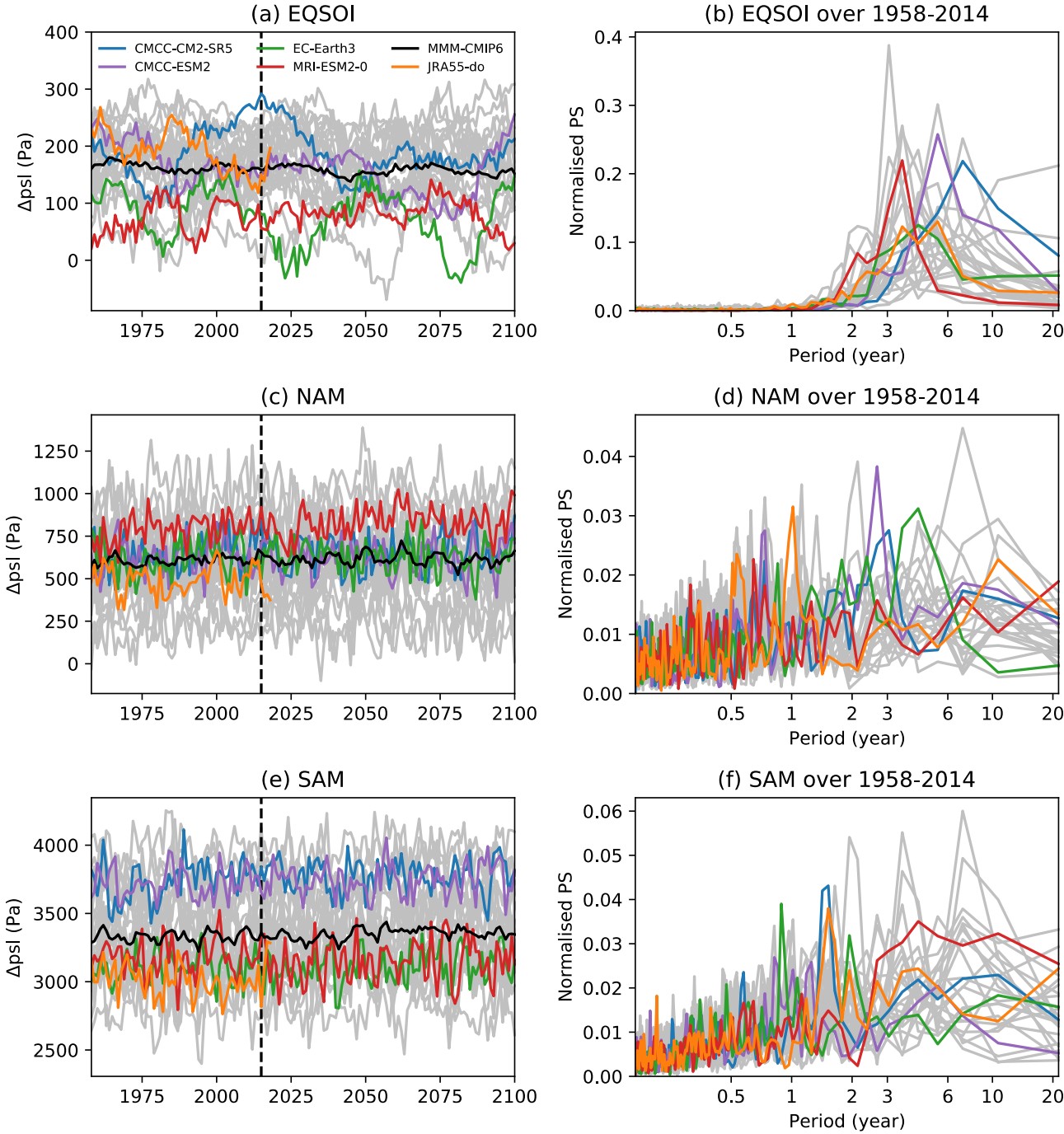

**Figure 4: Comparisons of major climate modes simulated by 26 CMIP6 models over the historical period (1958-2014) and the projection period under the SSP5-8.5 scenario (2015-2100).** Time series of annual mean sea level pressure difference between two regions representative of (a) EQSOI, (c) NAM, and (e) SAM. Normalised power spectra of (b) EQSOI, (d) NAM, and (f) SAM derived from time series of standardised monthly mean anomaly of sea level pressure difference over 1958-2014 based on Welch's method

(Welch, 1967). Blue, purple, green, and red denote 4 of the 24 models that provide the atmospheric output at the temporal resolutions needed for the projection experiment of IAMIP2. Gray denotes the remaining 22 models. Black denotes the multi-model mean of the 26 models. Orange denotes the JRA55-do dataset used for the atmospheric forcing for the historical experiment of IAMIP2.

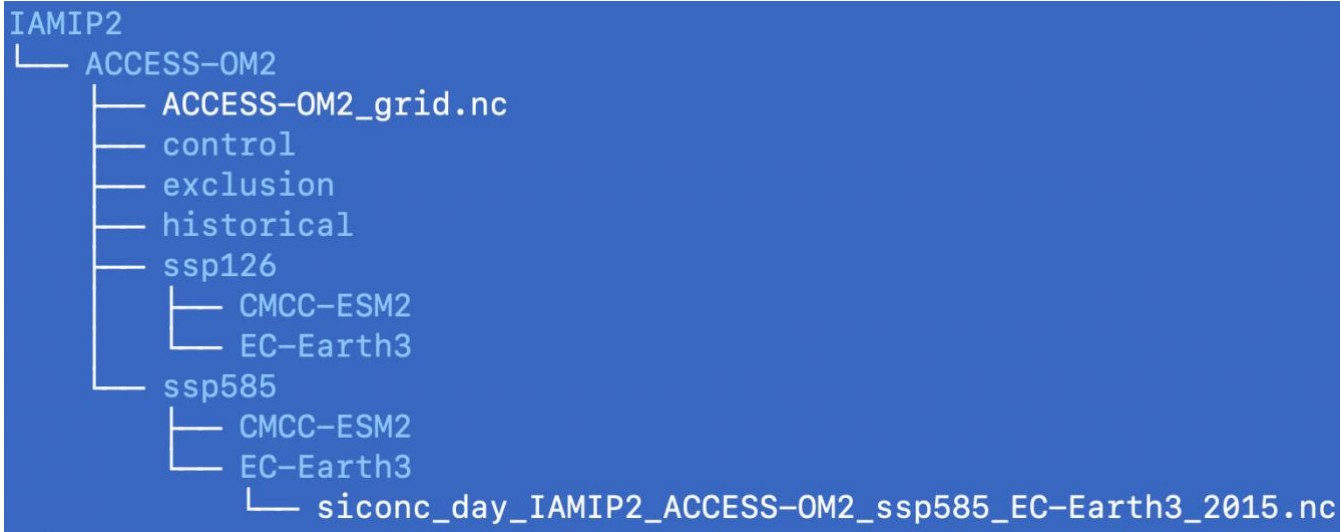

**Figure 5: An example file tree diagram for the IAMIP2 output.** Blue denotes the directories and white denotes files. A grid file (e.g., ACCESS-OM2_grid.nc) must be provided and placed under the model directory (e.g., IAMIP2/ACCESS-OM2/). Each diagnostic file (e.g., siconc_day_IAMIP2_ACCESS-OM2_ssp585_EC-Earth3_2015.nc) contains the output of a variable for a year.

**Table 1: List of participating models for IAMIP2.** For ocean and sea-ice ecosystem components, the letters denote nutrients (N), phytoplankton (P), zooplankton (Z), and detritus (D), which are followed by the numbers indicating the complexity. For example, N1 denotes one type of modelled nutrients (e.g., nitrate), whereas N2 indicates two types of modelled nutrients (e.g., nitrate and iron). The vertical resolution is rounded to the nearest integer.

| Model | ACCESS-OM2 | CESM-IARC | RASM | CanNEMO | NEMO-NAA | COCO-Arctic NEMURO |
|---|---|---|---|---|---|---|
| **Relevant CMIP6 model** | ACCESS-CM2, ACCESS-ESM1.5 | CESM2 | CESM2 | CanESM5 | CanESM5 | MIROC6 |
| **Ocean dynamics** | MOM5.1 | POP2 | POP2 | OPA | OPA | COCO4.9 |
| **Sea-ice dynamics** | CICE5.1.2 | CICE5.1.2 | CICE5.1.2 | LIM2 | LIM2 | COCO4.9 |
| **Ocean ecosystem** | N1P1Z1D1 | N4P3Z1D1 | N4P3Z1D1 | N3P2Z2D2 | N3P2Z2D2 | N3P2Z3D2 |
| **Sea-ice ecosystem** | N1P1 | N4P1 | N4P1 | N2P1 | N2P1 | N3P1D1 |
| **Spatial domain** | Global | Global | Pan-Arctic | Global | Pan-Arctic | Pan-Arctic |
| **Horizontal resolution** | 1° | 1° | 1/12° | 1° | 1/4° | 1/4° |
| **Vertical resolution of** | 2 m | 10 m | 5 m | 6 m | 1 m | 2 m |

| | | | | | |
|---|---|---|---|---|---|
| the surface ocean layer | | | | | |
| **Bottom-ice ecosystem layer thickness** | 0.03 m | 0.03 m | 0.03 m | 0.03 m | 0.02 m |
| **Reference** | (Jeffery et al., 2016; Kiss et al., 2020; Ziehn et al., 2020) | (Jin et al., 2018) | (Jin et al., 2018) | (Hayashida et al., 2019; Swart et al., 2019) | (Hayashida et al., 2019; Hu and Myers, 2013) | (Watanabe et al., 2015) |

**Table 2: List of model diagnostics for IAMIP2.** Where applicable, variable names and units follow the CMIP6 convention (https://earthsystemcog.org/projects/wip/CMIP6DataRequest). Tiers refer to mandatory (Tier 1) and optional (Tier 2).

| Variable name | Long name | Units | Tier |
|---|---|---|---|
| siconc | Percentage of grid cell covered by sea ice | % | 1 |
| sithick | Actual sea ice thickness (sea-ice volume divided by sea-ice-covered area) | m | 1 |
| sisnthick | Actual snow thickness (snow volume divided by snow-covered area) | m | 1 |
| intppbi | Vertically integrated primary organic carbon production by bottom ice algae | mol m$^{-2}$ s$^{-1}$ | 1 |
| intpp | Vertically integrated primary organic carbon production by phytoplankton | mol m$^{-2}$ s$^{-1}$ | 1 |
| epc100 | Downward flux of particulate organic carbon at 100 m | mol m$^{-2}$ s$^{-1}$ | 2 |
| phycbi | Bottom ice algae carbon concentration | mol m$^{-3}$ | 1 |
| phycos | Sea surface phytoplankton carbon concentration | mol m$^{-3}$ | 1 |
| no3bi | Bottom ice dissolved nitrate concentration | mol m$^{-3}$ | 1 |
| no3os | Sea surface dissolved nitrate concentration | mol m$^{-3}$ | 1 |
| parbi | Downwelling photosynthetic radiance flux at bottom ice | W m$^{-2}$ | 2 |
| paros | Downwelling photosynthetic radiance flux at sea surface | W m$^{-2}$ | 2 |
| sst | Sea surface temperature | °C | 1 |
| sss | Sea surface salinity | PSU | 1 |
| mlots2t | Ocean mixed layer thickness defined by sigma-t | m | 2 |
| dissicos | Sea surface dissolved inorganic carbon concentration | mol m$^{-3}$ | 2 |
| talkos | Sea surface total alkalinity | mol m$^{-3}$ | 2 |
| fgco2 | Sea surface downward flux of carbon dioxide | kg m$^{-2}$ s$^{-1}$ | 2 |