# Peer review of "Ice Algae Model Intercomparison Project phase 2 (IAMIP2)"

_Geoscientific Model Development, 2020_

## Referee Comment (RC1) · Anonymous Referee #1 · 18 Dec 2020

General comment: The manuscript "Ice Algae Model Intercomparison Project phase 2 (IAMIP2)" presents the experimental protocol of a new model intercomparison project with focus on sea ice algae and biogeochemistry. This model intercomparison builds upon a previous one (IAMIP1) which investigated variability in sea ice algae production on a season to decadal scale in the Arctic. New compared to the IAMIP1, IAMIP2 focuses on centennial scales and includes also the Antarctic region. In the present manuscript, the authors describe the coupled sea-ice—ocean—biogeochemical models that are, so far, taking part in the IAMIP2, they present the chosen atmospheric forcing dataset, and discuss the limitation of the IAMIP2 set up.

In the light of the fast changes experienced by polar regions as consequence of climate changes, and of the rapid decline of sea-ice cover, especially in the Arctic, it is of

extreme importance to understand the role that ice-produced carbon has for the polar marine ecosystems. Considering the limitations and constrains in sampling sea ice, the scientific community strongly relies on numerical experiments to quantify the important of sea ice algae and their role for the ecosystem on a large scale. This model intercomparison project, thus, is relevant for the community. The manuscript is well structured and the presentation of good quality. Figures and Table are appropriate. I recommend it for publication after the following minor edits are addressed by the authors.

Detailed comments: L30: add "together with phytoplankton"

L39: here the authors should acknowledge recent advancements in sampling the sea-ice biophysical properties on larger scales, the only reference to Miller et al., (2015) is not enough. See e.g., Lange et al., (2017); Castellani et al., (2020); Cimoli et al., (2020). Despite these recent advancements in characterizing sea ice algae spatial variability, we still rely on numerical models to obtain pan-Arctic and global estimates, thus stressing the relevance of such model intercomparison.

L271: add space between "40" and ""°C"

L315: add "especially in the Antarctic" before the full stop

[references] Lange B.A., Katlein C., Castellani G., Fernández-Méndez M., Nicolaus M., Peeken I. and Flores H. (2017) Characterizing Spatial Variability of Ice Algal Chlorophyll a and Net Primary Production between Sea Ice Habitats Using Horizontal Profiling Platforms. Front. Mar. Sci. 4:349. doi: 10.3389/fmars.2017.00349

Castellani G., Schaafsma F.L., Arndt S., Lange B.A., Peeken I., Ehrlich J., David C., Ricker R., Krumpen T., Hendricks S., Schwegmann S., Massicotte P. and Flores H. (2020) Large-Scale Variability of Physical and Biological Sea-Ice Properties in Polar Oceans. Front. Mar. Sci. 7:536. doi: 10.3389/fmars.2020.00536

Cimoli, E., Lucieer, V., Meiners, K.M. et al. Mapping the in situ microspatial distribution of ice algal biomass through hyperspectral imaging of sea-ice cores. Sci Rep 10, 21848

(2020). https://doi.org/10.1038/s41598-020-79084-6

---

## Referee Comment (RC2) · Anonymous Referee #2 · 7 Apr 2021

General comments: The manuscript "Ice Algae Model Intercomparison Project phase 2 (IAMIP2)." presents a well-designed and well-written protocol for a Model Intercomparison Project designed to address a knowledge gap in expected global and regional variability in sea ice algae abundance in the future. The authors describe the planned experiment design and rationale behind their choices of boundary conditions well, such as the reanalysis data and climate model output chosen for the forcing.

Overall, the proposed Modelling Intercomparison Project presents an opportunity for improving our understanding of the current ice algae communities and their expected development over the 21st century. I recommend this manuscript for publication after the following minor edits have been addressed. The tables and figures are appropriate.

Specific comments: As the authors mention in the manuscript, sampling of ice algae -related processes has been limited and is challenging, largely due to logistical constraints. As a result, also the knowledge of nutrient inventories in the polar regions includes significant uncertainties both for now and for the future (e.g. Lannuzel et. al 2020). I think that the approach of using the nutrient fields from World Ocean Atlas v2 (WOA13v2), following the approach chosen for the Ocean Model Intercomparison Project, is appropriate for this work. However, the availability of nutrients is one of the key uncertainties as a limiting factor for primary production in sea ice. The chosen assumption for future nutrient availability over the MIP period (unchanged from the WOA13v2 climatologies?) and resulting uncertainties for the output quantifications should be addressed in the manuscript, for example in the discussion.

L184: Please revise and clarify "zero velocity and sea level". Does this mean no major circulation patterns and currents in the atmosphere and the oceans, and a constant sea level everywhere despite varying sea level pressure?

L291: Revise the "...allows to quantify...". This goes back to my earlier comment. Accurate quantification the impact of ice algae on the ecosystems in the future would also require a good understanding of the nutrient availability in the future.

L325-327: please quantify the temporal resolutions needed to appropriately simulate the high-frequency climate variability or provide a reference

Technical corrections: L42: Consider revising the wording "unknown" to "unclear". This makes it sound like earlier contributions from IAMIP1 and other previous studies were rather insignificant

L47: Define IAMIP1 acronym here, where discussed for the first time

L147: The EVP acronym is not used elsewhere, is it necessary to be defined?

L176 & Fig.1 caption: Add "radiation" to "downward shortwave" and "downward longwave"

L307: Clarify what is meant with two-dimensional fields. Is all data with more dimensions disregarded in the archiving completely? If so, I would expect leaving out variables such as oceanic and atmospheric temperatures (3D) negatively affects reproducibility

L309: consider replacing the hyphen in "model's"

References: Lannuzel, D., Tedesco, L., van Leeuwe, M. et al. The future of Arctic sea-ice biogeochemistry and ice-associated ecosystems. Nat. Clim. Chang. 10, 983–992 (2020). https://doi.org/10.1038/s41558-020-00940-4

---

## Referee Comment (RC3) · Anonymous Referee #3 · 19 Apr 2021

This paper describes a model intercomparison project with the main goal of evaluating the influence of ice algae at regional and global scales. As the authors note, usually, ice algae are not included in Earth System Models (ESMs). In fact, none of the ESMs that are part of CMIP6 includes sea ice biogeochemistry. Therefore, it is necessary to know how important their inclusion in future ESMs might be. Whilst I fully agree about the importance of this experiment, there are several issues that I think authors should consider in revising this paper. This paper describes an experiment without presenting its results. The only "results" presented are about decisions regarding the atmospheric forcing used across all models included in the experiment and some other technicalities. Therefore, I wonder how this paper fits into the scope of a scientific journal. I could see it as a short comment but not so much as a full scientific paper at its present

stage. Looking into the Aims & Scope of Geoscientific Model Development (GMD) I can see that one of the possible types of manuscript is about "model experiment descriptions, including experimental details and project protocols". I am not sure if the idea behind this "model experiment descriptions" includes papers with only such descriptions and without corresponding results. So, I leave this up to the editor to decide. Now focusing on the experiment, itself. I think the idea is fine and extremely useful as I implied above. However, I think that some details should be addressed. For example, authors emphasize the evaluation of ice algae at regional and global scales. Their experiment is forced by atmospheric conditions from either a data assimilation product or ESMs. Their simulations do not include feedback from ice algae towards the atmosphere. Therefore, they cannot evaluate directly the influence of ice algae on atmospheric behavior. However, they can perhaps attempt to do so indirectly, through changes in carbon and DMS fluxes, dependent on including/excluding ice algae in their simulations. They can also evaluate the importance of ice algae on the ocean physical and biogeochemical processes, on the biological carbon pump and on the sink/source role of Arctic and Antarctic seas. I am sure authors know this quite well, but they should share it with the readers – specify the protocols they will use to evaluate the role of ice algae: which variables will be used from each model and what processes will they use to quantify their role at regional and global scales. Some models include dynamic elemental ratios, other do not include them but in the end, they will need to bring results to common currencies under some simplifying assumptions. In the text they merely write that the historical experiment is "designed to simulate changes in ice algae abundance and distribution", the "projection experiment is designed to simulate the projected changes in ice algae abundance and distribution", the "exclusion experiment is designed to simulate ocean biogeochemistry in the absence of ice algae" and the "control experiment is designed to diagnose artificial model drifts and to distinguish anthropogenic effects from natural variability". None of these sentences clarifies how the abundance and distribution of ice algae will be used to evaluate their regional and global significance. This should be clearly addressed

in the text. By the way, I suggest when referring to the various simulations that are part of the experiment to call them "simulations" or "treatments" and not "experiments" for the purposes of clarity. Whilst the overall experimental design seems fine, I see one aspect that I think authors should reconsider. When it comes to the projection simulation authors adopted the worst-case scenario - Shared Socioeconomic Pathway 5-8.5. Why this worst scenario which, probably, is not very likely to occur? The utility of this extreme scenario is a matter of debate (e.g. Hausfather and Peters, 2020). If I planned to use only one scenario, I think I would pick up the most likely one. The authors start the abstract by stating that "Ice algae play a fundamental role in shaping polar marine ecosystems and biogeochemistry". Moreover, at the beginning of the Introduction they write that "ice algae are the foundation of polar marine food webs". After these sentences one could ask: why then an experiment that attempts to evaluate the influence of ice algae at regional and global scales? Ice algae are quite likely foundation species for the ice-associated ecosystem, but I am not sure one may say they are foundation species for polar marine food webs "in general" before their quantitative contribution is compared to that of phytoplankton. Estimates of ice algal production are generally much lower than those of phytoplankton (see e.g. Jine at al. (2012) one of the co-authors of this study). These authors estimated an average Arctic ice-algal primary production of 21.7 Tg C year−1 for the period 1992–2007 corresponding to ∼5% of Arctic primary production. According to Arrigo et al. (2010) a limited number of studies have indicated that ice algal annual production is similar in the Arctic and Antarctic, ranging from 2 to 15 g C m−2 y−1 and from 0.3 to 38 g C m−2 y−1, and estimated to amount to 2–24% of total production in sea-ice covered marine areas (Arrigo et al., 2017). I am not implying that ice algae are not important, I am merely implying that they may be not so important as to call them the foundation of polar systems. A recent study by Kohlbach et al. (2021) suggest that in summer and winter pelagic calanoid copepods and amphipods rely much more on food from pelagic origin. These conclusions may differ if a similar study is conducted in spring-early summer, but they also emphasize the importance of pelagic food for

organisms that often are referred as strongly linked with the sea ice. When it comes to thermodynamics, Kauko et al. (2017) estimated melting rates of ∼1 cm per week for a refrozen lead in the drift ice of the Arctic Ocean, as a result of shortwave radiation absorbed by ice algae. Are these values relevant? Arctic-wide these correspond to a lot of ice of course but these values seem rather low when compared with melting rates resulting merely from physical processes. I guess that these arguments are by themselves a good reason for experiments like the one proposed in this paper, but we need their result to properly judge about the quantitative role of ice algae. The term "skeletal layer" is used in the text when referring to the bottom layer where most of ice algae are found in land-fast ice and that is considered in the models used in this experiment. I suggest using the term "bottom ice" or "bottom layer" and not skeletal layer which is present only during periods of ice growth (e.g. Hunke et al., 2015). I see a major limitation in this intercomparison project: it relies on models assuming ice algae only at bottom ice. Whereas this may be the dominant "picture" in land-fast ice, it does not seem to be the case in the much larger fraction of drift ice in the deep ocean, where often there is no bottom maximum of ice algal biomass. I quote here a sentence by Gradinger (1999) based on a study conducted in the Central Arctic and the Greenland Sea: "The lowermost 20 to 40 cm contained between 4 and 62% of the entire algal biomass. Consequently, ice biological studies, which deal only with the bottom few centimeters of the ice floes, will underestimate algal biomass and production by factors of up to 25". Studies by Melhikov based on the Sheba experiment and more recent studies based on the N-ICE2015 expedition confirm such findings. This issue is only slightly addressed in this paper, in the first paragraph of the Discussion where authors anticipate that "this limitation has a negligible effect on the estimation of depth-integrated biomass and primary production as demonstrated by field observations", quoting a study based on land-fast ice, when most of the ice in the Arctic and Antarctic is not land-fast. I finish this comment by merely emphasizing that the Los Alamos Sea Ice Model (used by some of the models included in this experiment) includes vertically resolved sea ice biogeochemistry besides the bottom

layer approach. I am aware of the cpu costs associated with its usage in comparison with a simpler approach, but this is no scientific reason to choose an approach which is, at least, "highly debatable". The authors justify the selection of ESMs atmospheric output based in part on whether they have or not the temporal resolutions needed for simulating high-frequency variability (e.g. line 209). However, I found no info about what is understood here as "high-frequency variability". I suggest specifying such details. I am not a native English speaker so I will not comment much on the language, which I think is clear. However, I have the impression that the text may be slightly improved, and I would recommend a revision by a native English speaker. I also find some repetitive sentences in various parts of the text about the rationale behind this experimental approach and its protocol that may be removed to avoid redundancies. In my view, if it is acceptable for GMD to publish a paper that merely describes an experiment, the acceptance of this paper should depend on the authors addressing the points above providing an in-depth description of the missing details and justifying convincingly the choice of the projection scenario. References Arrigo K.R., Mock T., Lizotte M.P., 2010. Primary producers and sea ice. In: David N. Thomas Gerhard S. Dieckmann (eds.). Wiley. https://doi.org/10.1002/9781444317145.ch8. Arrigo, K.R., 2017. Sea ice as a habitat for primary producers., In: Thomas, DN (ed.), Sea Ice, 3rd Edition, 52–369. Oxford, UK: Wiley-Blackwell. DOI: https://doi.org/10.1002/9781118778371.ch14. Gradinger, R., 1999. Vertical fine structure of the biomass and composition of algal communities in Arctic pack ice. Marine Biology 133: 745-754. Hausfather, Z., Peters, G.P., 2020. Emissions – the 'business as usual' story is misleading. Nature 577, 618-620. doi:https://doi.org/10.1038/d41586-020-00177-3. Hunke, E.C., Lipscomb, W.H., Turner, A.K., Jeffery, N., Elliot, S., 2015. CICE: the Los Alamos Sea Ice Model. Documentation and User's Manual Version 5. Jin, M., Deal, C., Lee, S.H., Elliot, S., Hunke, E., Maltrud, M., Jeffery, N., 2012. Investigation of Arctic sea ice and oceanic primary production for the period 1992–2007 using a 3-D global ice-ocean ecosystem model. Deep-Sea Res. II Top. Stud. Oceanogr. 81–84. Kauko, H. M., Taskjelle, T., Assmy, P., Pavlov, A. K., Mundy, C. J., Duarte, P.,

. . . Granskog, M. A. (2017). Windows in Arctic sea ice: Light transmission and ice algae in a refrozen lead. Journal of Geophysical Research-Biogeosciences, 122(6), 1486-1505. doi: 10.1002/2016JG003626. Kohlbach et al., 2021. Winter Carnivory and Diapause Counteract the Reliance on Ice Algae by Barents Sea Zooplankton. Front. Mar. Sci., 8:640050. doi: 10.3389/fmars.2021.640050.

Please also note the supplement to this comment:
https://gmd.copernicus.org/preprints/gmd-2020-305/gmd-2020-305-RC3-supplement.pdf

---

## Author Comment (AC1) · 26 Jul 2021

**26 July 2021**

**Dear Editor,**

**We provide our responses (in bold font) to the three reviewers' comments (in normal font) below. We hope that we have addressed these comments adequately for publication.**

**We note that Section 3.2 has been revised substantially as we found an issue in the earlier analysis of surface air temperature (Fig. 3), that spatial averaging was done accounting for both ocean and land grid cells. This led to averages that are biased towards temperatures over land especially for the Southern Hemisphere (south of 60S, which is mostly Antarctic continent). What is more appropriate for the scope of IAMIP2 is to average over ocean grid cells only because the values over land will be neglected in the sea ice-ocean model simulations. We realised this issue during the review process and hence took longer time for revisions.**

**Please let me know if there are any questions or concerns.**

**Best regards,**

**Hakase Hayashida, on behalf of the authors**

Anonymous Referee #1

General comment:

The manuscript "Ice Algae Model Intercomparison Project phase 2 (IAMIP2)" presents the experimental protocol of a new model intercomparison project with focus on sea ice algae and biogeochemistry. This model intercomparison builds upon a previous one (IAMIP1) which investigated variability in sea ice algae production on a season to decadal scale in the Arctic. New compared to the IAMIP1, IAMIP2 focuses on centennial scales and includes also the Antarctic region. In the present manuscript, the authors describe the coupled sea-iceâ˘Toceanâ ˘ A˘Tbiogeochemical ˘ models that are, so far, taking part in the IAMIP2, they present the chosen atmospheric forcing dataset, and discuss the limitation of the IAMIP2 set up. In the light of the fast changes experienced by polar regions as consequence of climate changes, and of the rapid decline of sea-ice cover, especially in the Arctic, it is of extreme importance to understand the role that ice-produced carbon has for the polar marine ecosystems. Considering the limitations and constrains in sampling sea ice, the scientific community strongly relies on numerical experiments to quantify the important of sea ice algae and their role for the ecosystem on a large scale. This model intercomparison project, thus, is relevant for the community. The manuscript is well structured and the presentation of good quality. Figures and Table are appropriate. I recommend it for publication after the following minor edits are addressed by the authors.
**We thank Anonymous Referee 1 for finding the importance of IAMIP2 for the polar marine community.**

Detailed comments:

L30: add "together with phytoplankton"
**Added as suggested.**

L39: here the authors should acknowledge recent advancements in sampling the seaice biophysical properties on larger scales, the only reference to Miller et al., (2015) is not enough. See e.g., Lange et al., (2017); Castellani et al., (2020); Cimoli et al., (2020). Despite these recent advancements in characterizing sea ice algae spatial variability, we still rely on numerical models to obtain pan-Arctic and global estimates, thus stressing the relevance of such model intercomparison.
**Thanks for these recent papers that demonstrate recent advancements in field measurements. We added the following text at the end of this sentence:**

**"(Miller et al., 2015), even though technological advancements in recent years have enabled field sampling at much larger scales (Castellani et al., 2020; Cimoli et al., 2020; Lange et al., 2017)."**

L271: add space between "40" and "∘C"
**Added as suggested.**

L315: add "especially in the Antarctic" before the full stop
**Added as suggested.**

[references]
Lange B.A., Katlein C., Castellani G., Fernández-Méndez M., Nicolaus M., Peeken I. and Flores H. (2017) Characterizing Spatial Variability of Ice Algal Chlorophyll a and Net Primary Production between Sea Ice Habitats Using Horizontal Profiling Platforms. Front. Mar. Sci. 4:349. doi: 10.3389/fmars.2017.00349
Castellani G., Schaafsma F.L., Arndt S., Lange B.A., Peeken I., Ehrlich J., David C., Ricker R., Krumpen T., Hendricks S., Schwegmann S., Massicotte P. and Flores H. (2020) Large-Scale Variability of Physical and Biological Sea-Ice Properties in Polar Oceans. Front. Mar. Sci. 7:536. doi: 10.3389/fmars.2020.00536
Cimoli, E., Lucieer, V., Meiners, K.M. et al. Mapping the in situ microspatial distribution of ice algal biomass through hyperspectral imaging of sea-ice cores. Sci Rep 10, 21848 (2020). https://doi.org/10.1038/s41598-020-79084-6

Anonymous Referee #2

General comments:

The manuscript "Ice Algae Model Intercomparison Project phase 2 (IAMIP2)." presents a well-designed and well-written protocol for a Model Intercomparison Project designed to address a knowledge gap in expected global and regional variability in sea ice algae abundance in the future. The authors describe the planned experiment design and rationale behind their choices of boundary conditions well, such as the reanalysis data and climate model output chosen for the forcing. Overall, the proposed Modelling Intercomparison

Project presents an opportunity for improving our understanding of the current ice algae communities and their expected development over the 21st century. I recommend this manuscript for publication after the following minor edits have been addressed. The tables and figures are appropriate.

**We thank Anonymous Referee 2 for finding our paper well-written and well-designed.**

Specific comments:

As the authors mention in the manuscript, sampling of ice algae -related processes has been limited and is challenging, largely due to logistical constraints. As a result, also the knowledge of nutrient inventories in the polar regions includes significant uncertainties both for now and for the future (e.g. Lannuzel et. al 2020). I think that the approach of using the nutrient fields from World Ocean Atlas v2 (WOA13v2), following the approach chosen for the Ocean Model Intercomparison Project, is appropriate for this work. However, the availability of nutrients is one of the key uncertainties as a limiting factor for primary production in sea ice. The chosen assumption for future nutrient availability over the MIP period (unchanged from the WOA13v2 climatologies?) and resulting uncertainties for the output quantifications should be addressed in the manuscript, for example in the discussion.

**Thanks for raising this important point. We agree that this will provide an additional uncertainty to our regional models whose boundary conditions will be prescribed by WOA13v2 in projection runs. We added this discussion to Section 5:**

**"One important source of biases in IAMIP2 projections by regional models is the lateral boundary conditions for nutrients. As a limiting factor for primary production in sea ice, prescribing the temporally-varying nutrient boundary conditions are desirable. However, this is not an easy task given the large uncertainty in projected changes in nutrients especially in polar regions (e.g., Lannuzel et al., 2020)."**

L184: Please revise and clarify "zero velocity and sea level". Does this mean no major circulation patterns and currents in the atmosphere and the oceans, and a constant sea level everywhere despite varying sea level pressure?

**Atmospheric circulation is not zero. Only the three-dimensional ocean current fields as well as sea level are set to zero in the very first time step. This is a common practice in ocean modelling (e.g. Griffies et al., 2016). Driven by the atmospheric forcing, major circulation patterns and currents in the upper ocean will spin up to quasi-steady states within a decade of simulations (e.g. Tsujino et al., 2020). We revised the text as follows for clarity:**

**"(three-dimensional oceanic velocity fields and two-dimensional sea level fields are all set to zero in the first time step)"**

**Griffies et al. (2016): OMIP contribution to CMIP6: experimental and diagnostic protocol for the physical component of the Ocean Model Intercomparison Project, GMD, https://doi.org/10.5194/gmd-9-3231-2016.**

**Tsujino et al. (2020): Evaluation of global ocean–sea-ice model simulations based on the experimental protocols of the Ocean Model Intercomparison Project phase 2 (OMIP-2), GMD, https://doi.org/10.5194/gmd-13-3643-2020.**

L291: Revise the "...allows to quantify...". This goes back to my earlier comment. Accurate quantification the impact of ice algae on the ecosystems in the future would also require a good understanding of the nutrient availability in the future.
**We have changed the time coverage of the exclusion experiment to the historical period (1958-2018), and therefore the uncertainty of nutrient availability in the future is not relevant anymore.**

L325-327: please quantify the temporal resolutions needed to appropriately simulate the high-frequency climate variability or provide a reference
**Added two references:**

**Holdsworth, A.M., and Myers, P.G. (2015). The Influence of High-Frequency Atmospheric Forcing on the Circulation and Deep Convection of the Labrador Sea. J. Clim. *28*, 4980–4996.**

**Lebeaupin Brossier, C., Béranger, K., and Drobinski, P. (2012). Sensitivity of the northwestern Mediterranean Sea coastal and thermohaline circulations simulated by the 1/12°-resolution ocean model NEMO-MED12 to the spatial and temporal resolution of atmospheric forcing. Ocean Model. *43–44*, 94–107.**

Technical corrections:

L42: Consider revising the wording "unknown" to "unclear". This makes it sound like earlier contributions from IAMIP1 and other previous studies were rather insignificant
**Revised.**

L47: Define IAMIP1 acronym here, where discussed for the first time
**Revised.**

L147: The EVP acronym is not used elsewhere, is it necessary to be defined?
**Deleted.**

L176 & Fig.1 caption: Add "radiation" to "downward shortwave" and "downward longwave"
**Added.**

L307: Clarify what is meant with two-dimensional fields. Is all data with more dimensions disregarded in the archiving completely? If so, I would expect leaving out variables such as oceanic and atmospheric temperatures (3D) negatively affects reproducibility
**Reproducibility is not an issue, because these diagnostics are the model output, and not the input. All input data including initial/boundary oceanic/atmospheric fields needed to reproduce the IAMIP2 experiments are publicly available as described in Section 2.**

**If one is interested in comparing subsurface oceanic properties to check for reproducibility, there are intpp and epc100 (Table 2) that can be used for that purpose.**

L309: consider replacing the hyphen in "model's"
**Replaced.**

References: Lannuzel, D., Tedesco, L., van Leeuwe, M. et al. The future of Arctic seaice biogeochemistry and ice-associated ecosystems. Nat. Clim. Chang. 10, 983–992 (2020). https://doi.org/10.1038/s41558-020-00940-4

Anonymous Referee #3

This paper describes a model intercomparison project with the main goal of evaluating the influence of ice algae at regional and global scales. As the authors note, usually, ice algae are not included in Earth System Models (ESMs). In fact, none of the ESMs that are part of CMIP6 includes sea ice biogeochemistry. Therefore, it is necessary to know how important their inclusion in future ESMs might be. Whilst I fully agree about the importance of this experiment, there are several issues that I think authors should consider in revising this paper. **We thank Anonymous Referee #3 for agreeing about the importance of IAMIP2. We have addressed your concerns below and revised the manuscript accordingly.**

This paper describes an experiment without presenting its results. The only "results" presented are about decisions regarding the atmospheric forcing used across all models included in the experiment and some other technicalities. Therefore, I wonder how this paper fits into the scope of a scientific journal. I could see it as a short comment but not so much as a full scientific paper at its present stage. Looking into the Aims & Scope of Geoscientific Model Development (GMD) I can see that one of the possible types of manuscript is about "model experiment descriptions, including experimental details and project protocols". I am not sure if the idea behind this "model experiment descriptions" includes papers with only such descriptions and without corresponding results. So, I leave this up to the editor to decide.
**These are two main reasons to publish the experimental design and protocol of IAMIP2. The first reason is to increase the visibility of IAMIP2, such that it reaches out to the community as much as possible. In fact, we now have an additional modelling group who have expressed an interest in participating IAMIP2 after our paper was published as preprint.**

**The second reason is to improve our methods by going through peer review process. Unlike simulations performed by a single model, it is difficult to re-do simulations by multiple models operated by different institutions, so we want to ensure that our methods are sound before performing simulations.**

**We believe that our paper fits into the scope of GMD. For reference, all of the MIPs of CMIP6 were published without the actual results of the experiments (https://gmd.copernicus.org/articles/special_issue590.html).**

Now focusing on the experiment, itself. I think the idea is fine and extremely useful as I implied above. However, I think that some details should be addressed. For example, authors emphasize the evaluation of ice algae at regional and global scales. Their experiment is forced by atmospheric conditions from either a data assimilation product or ESMs. Their simulations do not include feedback from ice algae towards the atmosphere. Therefore, they cannot evaluate directly the influence of ice algae on atmospheric behavior. However, they can perhaps attempt to do so indirectly, through changes in carbon and DMS fluxes, dependent on including/excluding ice algae in their simulations. They can also evaluate the

importance of ice algae on the ocean physical and biogeochemical processes, on the biological carbon pump and on the sink/source role of Arctic and Antarctic seas.

**Thanks for the detailed comments on the ways in which ice algae can influence the climate system at regional and global scales. The reviewer is correct that IAMIP2 cannot assess the influence directly, as there is no feedback from biogeochemistry to atmosphere. Our approach is to assess the potential influence, by quantifying carbon uptake (primary production), nutrient drawdown, biological carbon pump, etc., both at the regional (pan-Arctic/Antarctic) and global (sum of both hemispheres) scales. This can be a step toward implementing ice algae into the next generation of ESMs.**

I am sure authors know this quite well, but they should share it with the readers – specify the protocols they will use to evaluate the role of ice algae: which variables will be used from each model and what processes will they use to quantify their role at regional and global scales. Some models include dynamic elemental ratios, other do not include them but in the end, they will need to bring results to common currencies under some simplifying assumptions. In the text they merely write that the historical experiment is "designed to simulate changes in ice algae abundance and distribution", the "projection experiment is designed to simulate the projected changes in ice algae abundance and distribution", the "exclusion experiment is designed to simulate ocean biogeochemistry in the absence of ice algae" and the "control experiment is designed to diagnose artificial model drifts and to distinguish anthropogenic effects from natural variability". None of these sentences clarifies how the abundance and distribution of ice algae will be used to evaluate their regional and global significance. This should be clearly addressed in the text.

**Specification of which variables and processes used to quantify the role of ice algae can be inferred from Section 4, in which we describe the protocol for diagnostics. However, we can see that more explicit discussion will clarify the context. To address this issue, we added the following paragraph to Section 4:**

**"A few examples of potential use of these diagnostics for investigating various roles of ice algae are illustrated here. Their ecological role as the foundation of the polar marine food web and their relative importance can be quantified by comparing the biomass (*phycbi* and *phycos*; Table 2) and primary productivity between ice algae and phytoplankton (*intppbi* and *intpp*). The latter quantities can also be used to quantify the biogeochemical role of ice algae in carbon fixation and their contribution to the biological carbon pump can be assessed using particulate organic carbon export (*epc100*). A combination of physical (*siconc*, *paros*, *sst*, and *mlots2t*) and biogeochemical diagnostics (*no3os* and *phyos*) can be used to estimate dimethyl sulfide concentration (e.g., Bock et al., 2021; Galí et al., 2019) as well as its emission using the wind speed which is available as the atmospheric forcing fields (*uas* and *vas*; Figure 1)."**

By the way, I suggest when referring to the various simulations that are part of the experiment to call them "simulations" or "treatments" and not "experiments" for the purposes of clarity.

**Simulations are numerical experiments, so we don't think the term experiment needs to be changed to simulations. This paper is merely based on modelling, so it is clear that experiments refer to simulations.**

**Regardless, the word "numerical" is used to clarify that our experiments are numerical when introduced in Section 3 as well as in the Fig.2 caption.**

Whilst the overall experimental design seems fine, I see one aspect that I think authors should reconsider. When it comes to the projection simulation authors adopted the worst-case scenario - Shared Socioeconomic Pathway 5-8.5. Why this worst scenario which, probably, is not very likely to occur? The utility of this extreme scenario is a matter of debate (e.g. Hausfather and Peters, 2020). If I planned to use only one scenario, I think I would pick up the most likely one.

**We are aware of the debate such as Hausfather and Peters (2020). However, we are not entirely convinced that SSP5-8.5 is highly unlikely due to various carbon cycle mechanisms that are not accounted for in this and other previous studies that advocate for this claim (e.g., methane release from permafrost; see Michael Mann's blog post for further explanation; https://michaelmann.net/content/story-about-%E2%80%98business-usual%E2%80%99-story-misleading).**

**Another good reason for choosing SSP5-8.5 for the purpose of model intercomparison is that it will likely show the greatest difference among the models.**

**In the end, it is difficult to pick the 'most likely' one because we do not know the future. To account for this uncertainty in IAMIP2 projections, we have decided to conduct additional projection under the low emission scenario (SSP1-2.6). Accordingly, we revised the text in Section 3.2 as follows:**

**"The projection experiments are designed to simulate the projected changes in ice algae abundance and distribution throughout the twenty-first century under two of the greenhouse gas emission scenarios for CMIP6, known as the Shared Socioeconomic Pathways 1-2.6 and 5-8.5 (SSP1-2.6 and SSP5-8.5; O'Neill et al., 2016). SSP1-2.6 is a low emission scenario that informs the Paris Agreement goal of limiting the global warming to below 2 °C of the pre-industrial level. SSP5-8.5 is the highest emission scenario of CMIP6. Therefore, conducting these projections allows us to assess and compare between the impacts of strong mitigation and fossil-fueled development (O'Neill et al., 2016)."**

The authors start the abstract by stating that "Ice algae play a fundamental role in shaping polar marine ecosystems and biogeochemistry". Moreover, at the beginning of the Introduction they write that "ice algae are the foundation of polar marine food webs". After these sentences one could ask: why then an experiment that attempts to evaluate the influence of ice algae at regional and global scales? Ice algae are quite likely foundation species for the ice-associated ecosystem, but I am not sure one may say they are foundation species for polar marine food webs "in general" before their quantitative contribution is compared to that of phytoplankton. Estimates of ice algal production are generally much lower than those of phytoplankton (see e.g. Jine at al. (2012) one of the co-authors of this study). These authors estimated an average Arctic ice-algal primary production of 21.7 Tg C year$^{-1}$ for the period 1992–2007 corresponding to ~5% of Arctic primary production. According to Arrigo et al. (2010) a limited number of studies have indicated that ice algal annual production is similar in the Arctic and Antarctic, ranging from 2 to 15 g C m$^{-2}$ y$^{-1}$ and from 0.3 to 38 g C m$^{-2}$ y$^{-1}$, and estimated to amount to 2–24% of total production in sea-ice covered marine areas (Arrigo et al., 2017). I am not implying that ice algae are not important, I am merely implying that they may be not so important as to call them the foundation of polar systems. A recent study by Kohlbach et al. (2021) suggest that in summer and winter pelagic calanoid copepods and amphipods rely much more on food from pelagic origin. These conclusions may differ if a similar study is conducted in spring-early summer, but they also emphasize the importance

of pelagic food for organisms that often are referred as strongly linked with the sea ice. When it comes to thermodynamics, Kauko et al. (2017) estimated melting rates of ~1 cm per week for a refrozen lead in the drift ice of the Arctic Ocean, as a result of shortwave radiation absorbed by ice algae. Are these values relevant? Arctic-wide these correspond to a lot of ice of course but these values seem rather low when compared with melting rates resulting merely from physical processes. I guess that these arguments are by themselves a good reason for experiments like the one proposed in this paper, but we need their result to properly judge about the quantitative role of ice algae.

**We understand the reviewer's point here that the role of ice algae may have been overstated in the manuscript given their small quantitative contributions in the literature. We revised the first sentence in the abstract as follows:**

**"Ice algae play a fundamental role in shaping sea-ice-associated ecosystems and biogeochemistry."**

**We also added the word "Together with pelagic phytoplankton," at the beginning of Introduction (this was also the suggestion by Reviewer 1):**

**"Together with pelagic phytoplankton, microalgae that colonize sea ice are the foundation of polar marine food webs."**

The term "skeletal layer" is used in the text when referring to the bottom layer where most of ice algae are found in land-fast ice and that is considered in the models used in this experiment. I suggest using the term "bottom ice" or "bottom layer" and not skeletal layer which is present only during periods of ice growth (e.g. Hunke et al., 2015).
**Thanks for the insight. We have dropped the term "skeletal" from the manuscript, so they are now referred to as "bottom-ice layer".**

I see a major limitation in this intercomparison project: it relies on models assuming ice algae only at bottom ice. Whereas this may be the dominant "picture" in land-fast ice, it does not seem to be the case in the much larger fraction of drift ice in the deep ocean, where often there is no bottom maximum of ice algal biomass. I quote here a sentence by Gradinger (1999) based on a study conducted in the Central Arctic and the Greenland Sea: "The lowermost 20 to 40 cm contained between 4 and 62% of the entire algal biomass. Consequently, ice biological studies, which deal only with the bottom few centimeters of the ice floes, will underestimate algal biomass and production by factors of up to 25". Studies by Melhikov based on the Sheba experiment and more recent studies based on the N-ICE2015 expedition confirm such findings. This issue is only slightly addressed in this paper, in the first paragraph of the Discussion where authors anticipate that "this limitation has a negligible effect on the estimation of depth-integrated biomass and primary production as demonstrated by field observations", quoting a study based on land-fast ice, when most of the ice in the Arctic and Antarctic is not land-fast. I finish this comment by merely emphasizing that the Los Alamos Sea Ice Model (used by some of the models included in this experiment) includes vertically resolved sea ice biogeochemistry besides the bottom layer approach. I am aware of the cpu costs associated with its usage in comparison with a simpler approach, but this is no scientific reason to choose an approach which is, at least, "highly debatable".
**We agree that not accounting for sea-ice biogeochemistry above the bottom ice would underestimate the primary production by ice algae. However, we want to clarify that CICE5.1.2 adopted in ACCESS-OM2, CESM-IARC, and RASM does not provide an**

**option to resolve sea-ice biogeochemistry vertically. To the best of our knowledge, it is a feature that only became available from CICE6.**

**We revised the last sentence of the first paragraph of Section 5 (Discussion) as follows:**

**"We anticipate that this limitation has a negligible effect on the estimation of depth-integrated biomass and primary production in landfast sea ice (Meiners et al., 2018), but it could underestimate these quantities substantially for pack ice as demonstrated by field observations (Meiners et al., 2012). We consider this a necessary compromise in order to progress IAMIP2 with currently-available computational resources."**

The authors justify the selection of ESMs atmospheric output based in part on whether they have or not the temporal resolutions needed for simulating high-frequency variability (e.g. line 209). However, I found no info about what is understood here as "high-frequency variability". I suggest specifying such details.

**Here, high-frequency variability refers to time scales of ~daily. Previous studies demonstrate that high temporal resolution atmospheric forcing (<daily) is needed to simulate high-frequency (e.g., daily) oceanic variability (two references added to the revised manuscript, in response to Reviewer 2's comment above). We revised the manuscript to add this information:**

**"The IAMIP2 models are driven by the atmospheric output of selected CMIP6 models that provide the atmospheric forcing fields at the nominal spatial resolution of 100 km and at the temporal resolutions needed for simulating high-frequency (e.g., daily) variability (Holdsworth and Myers, 2015; Lebeaupin Brossier et al., 2012)."**

I am not a native English speaker so I will not comment much on the language, which I think is clear. However, I have the impression that the text may be slightly improved, and I would recommend a revision by a native English speaker. I also find some repetitive sentences in various parts of the text about the rationale behind this experimental approach and its protocol that may be removed to avoid redundancies.

**The manuscript has been reviewed internally multiple times by all co-authors, half of which are native English speakers. We believe that the language used in the revised manuscript is clear, as the reviewer stated. Regardless, the revised manuscript has been reviewed by all co-authors.**

In my view, if it is acceptable for GMD to publish a paper that merely describes an experiment, the acceptance of this paper should depend on the authors addressing the points above providing an in-depth description of the missing details and justifying convincingly the choice of the projection scenario.

**We believe that we have adequately responded to all the concerns raised by the reviewer.**

References Arrigo K.R., Mock T., Lizotte M.P., 2010. Primary producers and sea ice. In: David N. Thomas Gerhard S. Dieckmann (eds.). Wiley. https://doi.org/10.1002/9781444317145.ch8. Arrigo, K.R., 2017. Sea ice as a habitat for primary producers., In: Thomas, DN (ed.), Sea Ice, 3rd Edition, 52–369. Oxford, UK: Wiley-Blackwell. DOI: https://doi. org/10.1002/9781118778371.ch14. Gradinger, R., 1999. Vertical fine structure of the biomass and composition of algal communities in Arctic pack ice. Marine Biology 133: 745-754. Hausfather, Z., Peters, G.P., 2020. Emissions – the 'business

as usual' story is misleading. Nature 577, 618-620. doi:https://doi.org/10.1038/d41586-020-00177-3. Hunke, E.C., Lipscomb, W.H., Turner, A.K., Jeffery, N., Elliot, S., 2015. CICE: the Los Alamos Sea Ice Model. Documentation and User's Manual Version 5. Jin, M., Deal, C., Lee, S.H., Elliot, S., Hunke, E., Maltrud, M., Jeffery, N., 2012. Investigation of Arctic sea ice and oceanic primary production for the period 1992–2007 using a 3-D global ice-ocean ecosystem model. Deep-Sea Res. II Top. Stud. Oceanogr. 81–84. Kauko, H. M., Taskjelle, T., Assmy, P., Pavlov, A. K., Mundy, C. J., Duarte, P., . . . Granskog, M. A. (2017). Windows in Arctic sea ice: Light transmission and ice algae in a refrozen lead. Journal of Geophysical Research-Biogeosciences, 122(6), 1486-1505. doi: 10.1002/2016JG003626. Kohlbach et al., 2021. Winter Carnivory and Diapause Counteract the Reliance on Ice Algae by Barents Sea Zooplankton. Front. Mar. Sci., 8:640050. doi: 10.3389/fmars.2021.640050.